# Understanding usual care for patients with multimorbidity: baseline data from a cluster-randomised trial of the 3D intervention in primary care

Katherine Chaplin,[1] Peter Bower,[2] Mei-See Man,[1,3] Sara T Brookes,[3] Daisy Gaunt,[3] Bruce Guthrie,[4] Cindy Mann,[1] Stewart W Mercer,[5] Imran Rafi,[6] Alison R G Shaw,[1] Chris Salisbury[1]

For numbered affiliations see end of article.

**Correspondence to**
Professor Chris Salisbury;
c.salisbury@bristol.ac.uk

## ABSTRACT

**Objectives** Recent evidence has highlighted the high prevalence and impact of multimorbidity, but the evidence base for improving management is limited. We have tested a new complex intervention for multimorbidity (the 3D model). The paper describes the baseline characteristics of practices and patients in order to establish the external validity of trial participants. It also explores current 'usual primary care' for multimorbidity, against which the 3D intervention was tested.

**Design** Analysis of baseline data from patients in a cluster-randomised controlled trial and additional data from practice staff.

**Setting** Primary care in the UK.

**Participants** Patients with multimorbidity (n=5253) and 154 practice staff.

**Primary and secondary outcome measures** Using surveys and routinely available data, we compared the characteristics of participating and non-participating practices and participating and non-participating eligible patients. Baseline questionnaire data from patient participants was used to examine participant illness burden, treatment burden and perceptions of receiving patient-centred care. We obtained data about usual care preintervention from practice staff using questionnaires and a structured pro forma.

**Results** Participating practices were slightly larger, in less deprived areas, and with slightly higher scores for patient satisfaction compared with non-participating practices. Patients with dementia or learning difficulties were likely to be excluded by their general practitioners, but comparison of participants with non-participants identified only minor differences in characteristics, suggesting that the sample was otherwise representative. Patients reported substantial illness burden, and an important minority reported high treatment burden. Although patients reported relatively high levels of satisfaction with care, many reported not having received potentially important components of care.

**Conclusion** This trial achieved good levels of external validity. Although patients were generally satisfied with primary care services, there was significant room for improvement in important aspects of care for multimorbidity that are targeted by the 3D intervention.

## Strengths and limitations of this study

► Data on the external validity of trial populations are often not available, but recruitment using routine general practitioner records allowed us to compare participants and non-participants.

► We collected detailed data on care for multimorbidity using validated scales, complemented with data from staff for a more comprehensive assessment.

► Comparisons of participants and non-participants were limited to data available in routine records.

► Data on delivery and quality of care were generally based on patient and clinician self-report.

**Trial registration number** ISRCTN06180958; Post-results.

## INTRODUCTION

Recent evidence has highlighted the importance of multimorbidity for current health policy.[1] Multimorbidity among long-term conditions is the norm among older patients, and is common at a younger age in deprived populations.[2,3] It is associated with significant impacts on quality of life, mortality and healthcare utilisation.[1]

There is increasing consensus on the sort of care that is required for the management of patients with multimorbidity.[1,4,5] Much of this derives from consensus about high-quality care for long-term conditions more generally, with a focus on care planning, shared decision-making and self-management.[1,6–8] However, management of patients with multimorbidity also raises specific challenges, such as how to prioritise among conditions and how best to manage the treatment burden experienced due to multiple treatments and multiple appointments.[7,9] The increased prevalence of depression in multimorbidity is

well recognised, and comorbid depression is associated with worse outcomes.[10]

However, the evidence base for the management of multimorbidity remains sparse. A recent Cochrane review reported only 18 randomised trials specifically targeting multimorbidity, and concluded that 'there are remaining uncertainties about the effectiveness of interventions for people with multimorbidity in general due to the relatively small number of RCTs conducted in this area to date.[6 11] The National Institute for Health and Care Excellence (the leading UK organisation for the development of clinical guidelines) has published guidelines for the clinical assessment and management of multimorbidity, reviewing the evidence for varying 'format of encounters' in people with multimorbidity (including longer consultations, structured recall, involving the patient in agenda-setting and multiprofessional appointments) and for primary care-based comprehensive geriatric assessment.[1] However, the evidence available did not support any specific recommendation on how to organise primary care to better meet the needs of people with multimorbidity. Instead the guideline development group recommended that trials were needed evaluating new organisational approaches for people with multimorbidity.

The Cochrane review suggested that, given the complexity of needs and management of patients with multimorbidity, interventions are likely to be 'complex' (ie, 'involving several components acting in concert to improve care').[6 11] Our team has developed the 3D model for the management of multimorbidity in primary care. The model is described in full elsewhere,[12] and has recently undergone evaluation in a large-scale randomised controlled trial with concurrent economic and process evaluation.[12 13]

Key problems posed by current healthcare organisation and experienced by patients with multimorbidity are a lack of holistic patient-centred care, a high burden of illness and a high level of treatment burden due to multiple medications and the need to attend numerous appointments. Figure 1 shows how the 3D approach addresses these problems. The basis for 3D is the patient-centred care model[5 14 15] which includes four components:

► A focus on the patient's individual disease and illness experience.
► A biopsychosocial perspective.
► Finding common ground on what the problem is and mutually agreeing management plans.
► Enhancing the relationship between the patient and doctor (the therapeutic alliance).

The Medical Research Council has a well-developed framework for the development, evaluation and description of complex interventions.[16] Recent work in this area has also emphasised two additional issues. First, there is a need to understand the practice and patient populations who actually enter trials of complex interventions, compared with those who are potentially eligible, to better

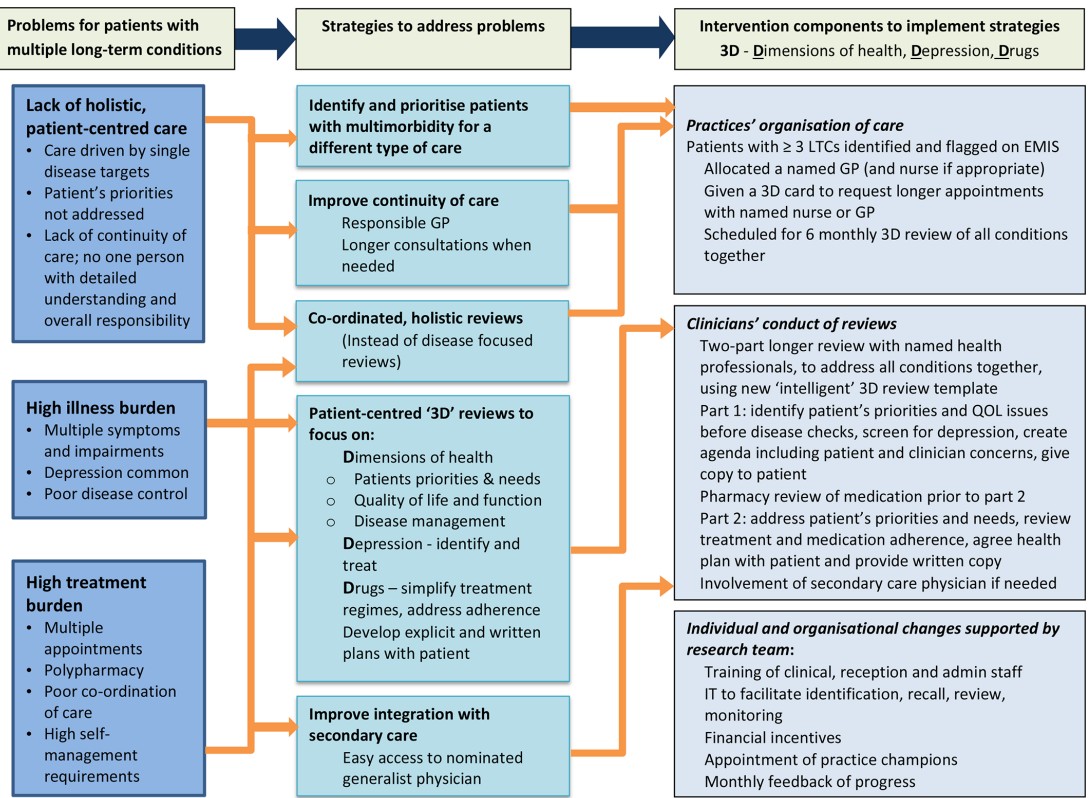

**Figure 1** 3D logic model including theoretical mechanisms of action. GP, general practitioner; LTCs, long-term conditions; QOL, quality of life.

understand the external validity of the study.[17 18] Second, there is a need to better understand the comparator to the intervention (in this case, 'usual primary care') in order to understand the content and quality of care against which the complex intervention is being tested.[19] The aims of this study are therefore to:

1. Compare practices and patients participating in the trial with non-participants.
2. Describe the characteristics of participating patients at baseline in terms of their experiences of (a) illness burden (b) treatment burden and (c) patient-centred care.
3. Describe usual care for people with multimorbidity.

## METHODS
### Design
The design of the 3D trial and process evaluation has been described in full[12 13] and is briefly summarised here. The 3D trial is a multicentre pragmatic, two-arm, practice-level cluster-randomised controlled trial. In the UK, each patient is registered to receive free healthcare under the National Health Service (NHS) from one local general practice. For most patients, chronic disease management is provided by general practitioners (GPs) and nurses within their registered practice with little or no involvement from hospital specialists. Practices are incentivised to provide high-quality care for many specified chronic diseases by the NHS Quality and Outcomes Framework (QOF) pay-for-performance scheme.[20]

The 3D study is based in general practices in three areas: Bristol and Greater Manchester in England and Ayrshire in Scotland. Volunteer practices were recruited from areas with a range of socioeconomic characteristics. For inclusion practices had to have at least 2 GPs and 4500 registered patients and to use the EMIS clinical IT system (used by the majority of practices in the UK). Inclusion criteria for patients were age 18+ and having three or more types of long-term condition from those included in the QOF (online supplementary appendix A). We decided to include patients with three or more (rather than two or more) conditions in order to focus effort on more complex patients who may have more to gain from a new model of organisation. Up to 150 potentially eligible patients were randomly selected from each practice by a researcher using a random number function in Microsoft Excel, and using an anonymous patient identifier. Selected patients' notes were screened by their GPs against the following exclusion criteria: having a life expectancy of less than 12 months; serious suicidal risk; known to be leaving the practice within 12 months; unable to complete questionnaires in English even with the help of carers; actively taking part in other research involving extra visits to primary care or other health services; lacking capacity to consent (Scotland only) or being considered unsuitable for the research study by their GP. All remaining patients were sent an invitation from their practice including information about

the study, a consent form and baseline questionnaire. Patients self-consented by returning the consent form and completed baseline questionnaire to the research team, using a freepost envelope. Non-respondents were sent one postal reminder 10–14 days later, supplemented by a telephone reminder in those practices where recruitment targets were not met.

### Patient data
We had data on two groups of patients. For patients who were invited to the trial ('potentially eligible patients'), we had data on age, sex and QOF-recorded conditions.

For eligible patients who consented to take part ('participating patients'), data were also available from the baseline questionnaire measuring depression (Hospital Anxiety and Depression Scale, HADS),[21] quality of life (EQ5D-5L),[22] illness burden (Bayliss *et al*),[23] treatment burden Multimorbidity Treatment Burden Questionnaire (MTBQ),[24] patients' perception of the quality of chronic illness care (Patient Assessment of Care in Chronic Illness Care (PACIC))[25] and perceived empathy of GPs and nurses (Consultation and Relational Empathy (CARE) measure).[26]

The patient questionnaire included several questions about holistic patient-centred care. These included the PACIC measure and the CARE measure, along with two questions from the Long-Term Conditions 6 (LTC6) questionnaire.[27] Three further questions were included regarding satisfaction with current care, whether patients usually saw their preferred GP, and whether they had a written care plan (all based on the national GP Patient Survey).[28]

### Staff perceptions and practice data on the organisation of care
At the start of the trial, participating GPs and practice nurses completed a purpose-designed questionnaire about their beliefs and attitudes regarding care of patients with multimorbidity. Researchers training the nurses and GPs in intervention practices asked them to complete the questionnaire before the training began. In usual care practices, the questionnaire was distributed via the practice manager and followed up with one researcher reminder where there was a poor response. The questionnaire consisted of 12 statements that were scored from 1 ('strongly disagree') to 5 ('strongly agree') (online supplementary appendix B). Only those questions which can be compared with patient's perspectives have been reported.

In addition, information about how the practice organised usual care for patients with long-term conditions was collected from all practices through a structured pro forma completed by a single key respondent in each practice (usually the practice manager) via an emailed survey supplemented by telephone or face-to-face interview. This covered staff resources, organisation of long-term condition review clinics and practice

policy on medication reviews, care plans and continuity of care.

## Analysis

In order to compare practices and patients participating in the trial with non-participants, we compared the characteristics of practices in the 3D trial with practices in the same Clinical Commissioning Group and national data. We assessed differences in patient populations (age, deprivation), practice size and published assessments of quality (the percentage of targets met within the QOF)[29] and patient satisfaction (based on the national GP Patient Survey).[28]

We described the demographic and clinical characteristics of patients at each stage of recruitment to 3D—those identified as potentially eligible but excluded by their GP, those eligible but not participating (due to non-response or actively declining), and those who agreed to participate in the study. All comparisons were analysed in multilevel regression models which included practice as a random effect. For participants in the trial, we present descriptive data on patients self-reported baseline measures of their illness burden and treatment burden. Because the number of potential participants is large (n=5253), we have interpreted whether absolute differences are meaningful rather than relying only on p values (since even small and non-meaningful differences will generate small p values with large samples).

To describe the extent to which current care for people with multimorbidity is patient centred from the perspective of patients we present participant responses to individual question items from the baseline patient questionnaire reflecting key concepts in patient-centred care.

We also provide data about staff views about care for people with multimorbidity and report descriptive data from the structured pro forma about usual care for patients with multimorbidity in all practices participating in the trial.

For all descriptive analyses we have calculated intracluster correlation coefficients (ICCs) to demonstrate the extent of practice-level variability. All analyses were performed using Stata V.15 (StataCorp).

## Patient and public involvement

An active group of up to 14 patients and carers provided a service user perspective. Through regular meetings with the research team, they contributed to refinement of the research questions and design of the intervention. The group were consulted about the perceived burden of the intervention, and provided valuable feedback on the specific outcome measures chosen including helping to develop the measure of treatment burden. They particularly contributed to communications with participants in recruitment materials and regular newsletters about progress. The findings will be available to participants and the public on the trial website.

## RESULTS

### What types of practices and patients participated in the 3D trial, and how did they compare to non-participants?

Across the three sites, 68 practices expressed initial interest in the study, of which 35 signed up to the study. Two practices subsequently withdrew prior to randomisation. The remaining 33 practices (49% of those expressing interest) were randomised, 16 into the intervention arm and 17 to usual care. Descriptive characteristics of the 33 practices are shown in table 1. Compared with all practices in their local area, practices which agreed to participate tended to be slightly larger, in less deprived areas and had slightly higher scores for patient satisfaction (table 1).

The flow of patients into the trial is shown in figure 2. Between 20 May 2015 and 31 December 2015, a total of 9772 patients were identified as potentially eligible, representing 3.9% of the adult population. Of these, 5253 were randomly sampled from practice registers. GPs excluded 575 (11%) of those based on medical record data because they were ineligible or the GP felt it would be inappropriate to invite them to participate. Potential participants who were excluded by their GPs were more likely to have dementia or learning difficulties and less likely to have diabetes or respiratory conditions than those not excluded (table 2). There was considerable variation between practices in the percentage of patients excluded (mean=11.01%, SD=8.01%).

Of 4678 patients invited to participate, 1546 (33%) provided consent. Differences between participants and non-participants in terms of their health conditions were small, except that participants were less likely to have dementia than non-participants. Of the 11 types of long-term condition, which made people eligible for the trial the most commonly reported were cardiovascular disease (including hypertension, peripheral artery disease, chronic kidney disease, coronary heart disease and heart failure; affecting 93% of participants), diabetes (52%) and respiratory conditions (asthma or chronic obstructive pulmonary disease; 50%).

Baseline demographic and health data on excluded patients, non-participants and participants are shown in table 2. Excluded patients were more likely to be female, older and have four or more conditions than those invited. Participants and non-participants had very similar demographic characteristics and experienced a similar number of health conditions.

Two-thirds of patients (66%) reported having fair or poor health, with less than 7% reporting very good or excellent health (table 3). Although inclusion to the trial was based on QOF conditions in medical records, patients self-reported an average of seven conditions from the more comprehensive list included in the Bayliss measure.[23] Based on the HADS measure, more than one-third of patients (38%) reported anxiety or depression of at least mild severity.

On average, patients reported regularly taking 8 medications with 32% of patients taking at least 10 regular

**Table 1** Characteristics of participating and non-participating practices

| | Participating practices: Bristol (n=12) | Non-participating practices: BNSSG CCGs (n=86) | Participating practices: Manchester (n=11) | Non-participating practices: Manchester CCGs* (n=181) | All practices: England (n=7674) | Participating practices: Ayrshire and Arran (n=10) | Non-participating practices: Ayrshire and Arran (n=46) | All practices: Scotland (n=982) |
|---|---|---|---|---|---|---|---|---|
| **Size[39 40]** | | | | | | | | |
| Average list size (SD) | 11 360 (3950) | 9337 (3792) | 8531 (3768) | 6389 (3861) | 7450 (NR)† | 6874 (2813) | 6869 (3565) | 5736 (3591) |
| **Age profiles[40 41]** | | | | | | | | |
| % aged 65–74 | 10.3 | 8.7 | 12.1 | 10.9 | 17.2 | 12.4 | 12.1 | 10.2% |
| % aged 75–84 | 5.8 | 5.3 | 6.9 | 6.1 | 7.8 | 7.0 | 6.9 | 5.8% |
| % aged 85+ | 2.6 | 2.3 | 2.9 | 2.2 | 2.3 | 2.6 | 2.2 | 2.0% |
| **Deprivation[41 42]** | | | | | | | | |
| Deprivation, mean (SD) | 17.3 (13.0) | 20.0 (11.3) | 14.9 (8.3) | 26.5 (11.5) | 21.5 | 28.8 (14.9) | 32.5 (15.5) | |
| **QOF[29 43]** | | | | | | | | |
| QOF achievement (2014/2015) (%) | 98.7 | 96.6 | 96.2 | 96.7 | 95.5 | 99.8 | 98.8 | 97.3 |
| **Satisfaction with GP surgery[44 45] (%)** | | | | | | | | |
| Very positive (%) | 46.4 | 41.9 | 50.0 | 51.3 | 43 | 49.1 | 47 | 87% |
| Positive (%) | 42.4 | 44.2 | 39.6 | 36.8 | 42 | 39.2 | 39 | |
| Neutral (%) | 8.3 | 9.4 | 7.0 | 8.1 | 10 | 9.8 | 12 | 10% |
| Negative (%) | 2.9 | 4.5 | 3.5 | 3.8 | 5 | 1.9 | 2 | 3% |

*Eastern Cheshire, South Cheshire, St Helens, Wigan and Wirral.
†Deprivation is based on IMD 2010 for England and SIMD 2012 for Scotland.
BNSSG, Bristol, North Somerset, South Gloucestershire; CCGs, Clinical Commissioning Groups; GP, general practitioner; IMD, Index of Multiple Deprivation; QOF, Quality and Outcomes Framework; SMID, Scottish Index of Multiple Deprivation.

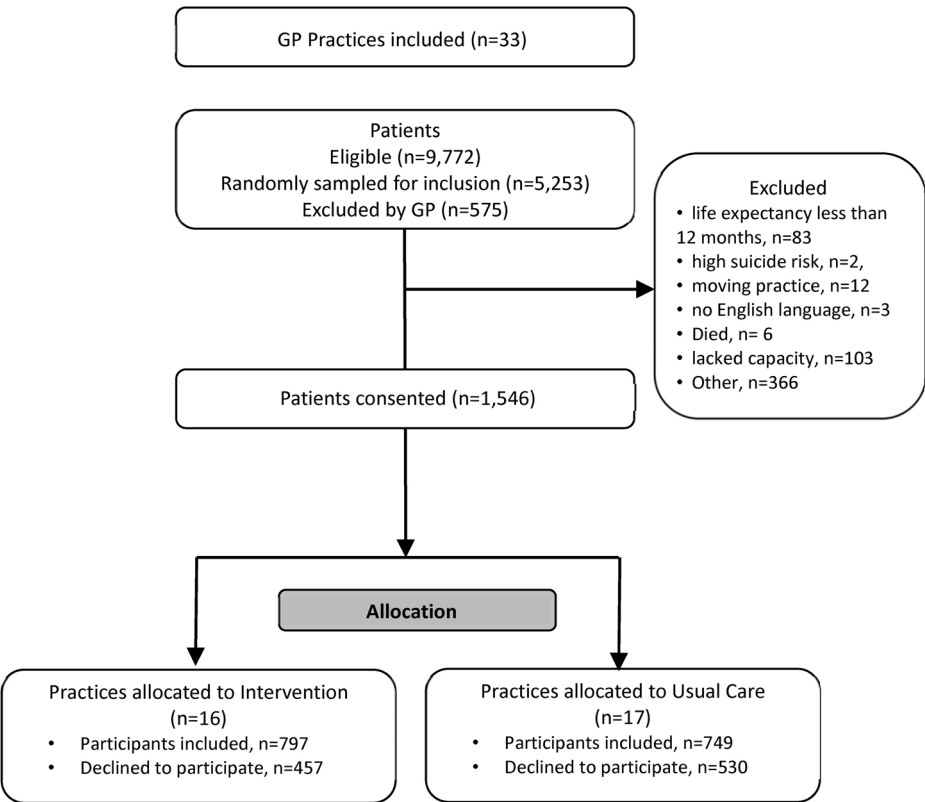

**Figure 2** Flow of patients into the 3D trial. GP, general practitioner.

medications (table 4). More than half (55%) reported at least a moderate level of treatment burden, with a score of at least 10 on the MTBQ. This score would be achieved, for instance, by having some difficulty in at least two areas of healthcare, or severe difficulty in at least one area.

Table 4 shows that most patients indicated that a GP or primary care nurse was responsible for their long-term condition, and reported relatively high levels of overall satisfaction with their care, although reported levels of care coordination were somewhat lower. Three-quarters had a preferred GP and of these 66% saw that GP 'most of the time'. In terms of 'whole person care', approximately two-thirds of patients reported that their GP and nurse were 'excellent' or 'very good' at 'being interested in them as a whole person'. However, only 37% reported that their care was always 'joined up'.

The data show that many patients do not perceive care as patient centred in terms of focusing on an individual's experience and agreeing management plans. A relatively high proportion of patients (35%) reported 'rarely' or 'not at all' discussing what was most important to them in terms of their health (table 4). Only 10% of participants reported having a care plan. Scores on the PACIC scale were around the midpoint of the scale, with the highest ratings for 'activation' and 'decision support', and the lowest for 'goal setting' and 'follow-up' (table 4).

### The extent to which current care for people with multimorbidity is patient centred from the perspective of primary care clinicians

The vast majority (88%) of clinicians agreed that patients with multimorbidity have a special need for patient-centred care and over 95% agreed that continuity of care improves patient-centred care (table 5). Most clinicians agreed that patients with a long-term condition should be given a care plan and that they were more likely to adhere to goals they had suggested themselves, but were evenly divided on whether patients preferred the clinician to make the plan. More than half of the clinicians agreed that patients' main concerns may be overlooked in long-term condition reviews (table 5). Almost all clinicians (93%) felt that patients with multimorbidity need longer appointments to address all their concerns.

Only 1 of the 33 practices said they routinely provided patients with a written care plan (and 80% of the patients in that practice said they did not have a written care plan). Only one-third (n=10) of practices had an active policy to encourage continuity of care, with the majority of others saying they try to accommodate patient preference. Only 36% of practices said they routinely performed depression screening while 76% said they conducted face-to-face medication reviews at least annually. All except two practices said they tried to combine reviews of some long-term conditions which might lessen treatment burden and improve joined up care (table 6).

**Table 2** Comparison of participating and non-participating patients (long-term conditions on QOF registers, demographic and clinical characteristics)*

| | Excluded† (n=575) | Non-participants‡ (n=3132) | Participants (n=1546) | Excluded versus invited§¶ | Difference between participants and non-participants¶ |
|---|---|---|---|---|---|
| Dementia | 225 (39%) | 340 (11%) | 60 (4%) | OR=0.12, p<0.001 | −7% OR=0.32, p<0.001 |
| Depression | 246 (43%) | 1250 (40%) | 559 (36%) | OR=0.83, p=0.037 | −4% OR=0.87, p=0.037 |
| Severe mental health group | 47 (8%) | 200 (6%) | 66 (4%) | OR=0.66, p=0.014 | −2% OR=0.66, p=0.004 |
| Learning difficulties | 48 (8%) | 84 (3%) | 14 (1%) | OR=0.22, p<0.001 | −2% OR=0.33, p<0.001 |
| Epilepsy | 46 (8%) | 185 (6%) | 76 (5%) | OR=0.68, p=0.021 | −1% OR=0.81, p=0.128 |
| Diabetes | 198 (34%) | 1613 (52%) | 812 (53%) | OR=2.07, p<0.001 | 0% OR=1.03, p=0.641 |
| Cardiovascular disease group** | 521 (91%) | 2875 (92%) | 1445 (93%) | OR=1.30, p=0.091 | +1% OR=1.25, p=0.066 |
| Stroke or TIA | 215 (37%) | 1050 (34%) | 527 (34%) | OR=0.87, p=0.124 | +1% OR=1.02, p=0.741 |
| Rheumatoid arthritis | 37 (6%) | 196 (6%) | 103 (7%) | OR=0.99, p=0.964 | +1% OR=1.06, p=0.631 |
| Respiratory (asthma or COPD) | 191 (33%) | 1456 (46%) | 770 (50%) | OR=1.87, p<0.001 | +4% OR=1.21, p=0.003 |
| Atrial fibrillation | 164 (29%) | 928 (30%) | 530 (34%) | OR=1.17, p=0.114 | +4% OR=1.19, p=0.009 |
| Male | 242 (42%) | 1452 (46%) | 763 (49%) | OR=0.81, p=0.018 | OR=0.90, p=0.078 |
| Age, mean, (SD), range | 77.14 (14.2) 18–106 | 71.35 (13.4) 20–101 | 70.79 (11.5) 25–96 | β=−6.04, p<0.001 | β=−1.11, p=0.005 |
| Morbidity count, mean (SD), range | 3.39 (0.64) 3–6 | 3.26 (0.53) 3–7 | 3.23 (0.48) 3–6 | β=−0.14, p<0.001 | β=−0.03, p=0.044 |
| Three comorbidities | 395 (69%) | 2444 (78%) | 1234 (80%) | OR=1.66, p<0.001†† | OR=1.12, p=0.148†† |
| Four comorbidities | 140 (24%) | 577 (18%) | 277 (18%) | | |
| Five comorbidities | 35 (6%) | 99 (3%) | 31 (2%) | | |
| Six comorbidities | 5 (1%) | 11 (0.4%) | 4 (0.3%) | | |
| Seven comorbidities | | 1 (0.03%) | | | |

Baseline characteristics of participating patients in terms of (1) illness burden and (2) treatment burden.
*Since an inclusion criterion was having three or more conditions, the percentages in each column exceed 100%.
†Eligible on record search but excluded by GP before invitation.
‡Non-participants combine patients who declined and those who did not respond.
§Invited includes non-participants and participants combined.
¶ORs were calculated using a multilevel logistic regression with practice included as a random effect, β coefficients were calculated using a multilevel linear regression with practice included as a random effect.
**Includes hypertension, peripheral artery disease, chronic kidney disease, coronary heart disease and/or heart failure.
††3 comorbidities versus 4–7 comorbidities.
COPD, chronic obstructive pulmonary disease; GP, general practitioner; QOF, Quality and Outcomes Framework; TIA, Transient Ischaemic Attack.

The ICCs reported in tables 3–6 suggest low levels of clustering (ICC <0.05) for all outcomes except for some variables relating to practice organisation of care, such as whether care is provided mainly by nurses or doctors, participant satisfaction with care, and clinicians' attitudes to written care plans.

## DISCUSSION
### Summary of the findings
The paper describes usual care for people with high levels of multimorbidity using baseline data from a cohort of patients entering a trial. Comparison of patients entering the trial with non-participants identified only minor

**Table 3** Baseline data on illness and treatment burden (participants)

| | | | ICC (95% CI) |
|---|---|---|---|
| Health-related quality of life—EQ5D (n=1546) | Mean (SD), range –0.51 to 1.00 | 0.558 (0.287) | 0.033 (0.007 to 0.059) |
| General health (n=1546) | Poor | 321 (21%) | 0.034 (0.008 to 0.060) |
| | Fair | 681 (45%) | |
| | Good | 429 (28%) | |
| | Very good | 88 (6%) | |
| | Excellent | 5 (0.3%) | |
| Bayliss et al[23] | Mean count (SD), n, range 1–73 | 7.5 (3.2), 1543 | 0.003 (0.000 to 0.014) |
| | Mean illness burden* (SD), n, range 1–26 | 18.8 (12.4), 1458 | 0.023 (0.001 to 0.046) |
| Depression—HADS (n=1512) | Normal (0–7) | 932 (62%) | 0.041 (0.011 to 0.070) |
| | Mild (8–10) | 285 (19%) | |
| | Moderate (11–14) | 211 (14%) | |
| | Severe (15–21) | 84 (6%) | |
| Anxiety—HADS (n=1506) | Normal (0–7) | 964 (64%) | 0.029 (0.005 to 0.053) |
| | Mild (8–10) | 246 (16%) | |
| | Moderate (11–14) | 204 (14%) | |
| | Severe (15–21) | 92 (6%) | |
| Mean no of medications (self-report) | Mean (SD), range 0–34 | 8.36 (3.94) | 0.018 (0.000 to 0.039) |
| No of medications (n=1396) | 0–4 | 171 (12%) | 0.018 (0.000 to 0.039) |
| | 5–9 | 781 (56%) | |
| | 10–14 | 350 (25%) | |
| | 15+ | 94 (7%) | |
| Multimorbidity Treatment Burden Questionnaire (n=1524) | None (0) | 308 (20%) | 0.026 (0.003 to 0.049) |
| | Low (<10) | 385 (25%) | |
| | Medium (10–22) | 425 (28%) | |
| | High (≥22) | 406 (27%) | |

*Each self-reported health condition is weighted by the extent to which it affects the participant's life, from 1 (not at all) to 5 (a lot).
HADS, Hospital Anxiety and Depression Scale; ICC, intracluster correlation coefficient.

differences in demographic and clinical characteristics, suggesting good external validity. As anticipated, participants in the trial reported high levels of illness burden and treatment burden. Although participants reported relatively high levels of satisfaction with their relationships with professionals, responses to specific questions identified important gaps in their experience of care as patient centred. Although clinicians supported aspects of patient-centred care such as continuity of care and care plans, and claimed to provide these, the experiences of patients were variable. The results of this study suggest that there is significant room for improvement in many aspects of care for multimorbidity that are targeted by the 3D intervention. In particular, the results suggest a need for improvements in the continuity and coordination of care, more focus on the problems which matter most to patients (including mental as well as physical health),

more effort to reduce the burden of treatment and more attention to goal setting and sharing written care plans.

### Strengths and limitations
A key strength of this study was our ability to collect comparative data on 'potentially eligible' patients, to allow us to compare participants and non-participants. Data on the external validity of trial populations are often not available, but recruitment using routine GP records does provide significant advantages in this regard. We also collected detailed data on care for multimorbidity using validated scales and complemented these with data from staff to provide a more comprehensive assessment.

Detailed comparisons of participants and non-participants are inevitably difficult because more detailed survey data are by definition not available for non-participants, and comparisons are restricted to basic demographic

**Table 4** Baseline self-reported data on 'holistic patient-centred care'

| | Response | n(%) or mean (SD) | ICC (95% CI) |
|---|---|---|---|
| **Long-term condition care** | | | |
| Who manages your long-term conditions? (n=1436) | GP | 920 (64%) | 0.036 (0.008 to 0.064) |
| | Nurse | 361 (25%) | 0.056 (0.019 to 0.093) |
| | Matron | 8 (0.6%) | 0.001 (0.000 to 0.013) |
| | Hospital doctor | 103 (7%) | 0.006 (0.000 to 0.021) |
| | Hospital nurse | 17 (1%) | 0.000 (0.000 to 0.012) |
| How satisfied are you with the care you get at your GP surgery? (n=1494) | Very dissatisfied | 36 (2%) | 0.067 (0.026 to 0.108) |
| | Fairly dissatisfied | 61 (4%) | |
| | Neither | 149 (10%) | |
| | Fairly satisfied | 489 (33%) | |
| | Very satisfied | 759 (51%) | |
| Do you think the support and care you receive is joined up and working for you? (n=1479)* | Not at all | 174 (12%) | 0.041 (0.011 to 0.071) |
| | Rarely | 165 (11%) | |
| | Some of the time | 590 (40%) | |
| | Always | 550 (37%) | |
| PACIC total (n=1232)† | Mean (SD), range 0–5 | 2.5 (1.0) | 0.044 (0.011 to 0.078) |
| Patient activation (n=1454)† | Mean (SD), range 1–5 | 3.0 (1.2) | 0.041 (0.011 to 0.070) |
| Decision support (n=1452)† | Mean (SD), range 1–5 | 2.9 (1.0) | 0.029 (0.005 to 0.054) |
| Goal setting (n=1443)† | Mean (SD), range 1–5 | 2.3 (1.1) | 0.029 (0.004 to 0.053) |
| Problem solving (n=1445)† | Mean (SD), range 1–5 | 2.7 (1.2) | 0.041 (0.011 to 0.072) |
| Follow-up/coordination (n=1432)† | Mean (SD), range 1–5 | 2.2 (1.0) | 0.035 (0.007 to 0.062) |
| **Continuity of care** | | | |
| Do you have a preferred GP? (n=1522) | Yes | 1148 (75%) | 0.038 (0.010 to 0.065) |
| If yes, how frequently do you see your preferred GP? (n=1141) | Always | 508 (44.5%) | 0.127 (0.062 to 0.192) |
| | A lot | 246 (21.5%) | |
| | Some | 294 (26%) | |
| | Never | 81 (7%) | |
| | Not tried | 8 (0.7%) | |
| Asked how my consultations with other doctors going (n=1399)† | Almost never/generally not | 888 (63%) | 0.037 (0.008 to 0.064) |
| | Sometime | 229 (16%) | |
| | Most of time/almost always | 282 (20%) | |
| **Whole person care** | | | |
| GP being interested in you as a whole person (n=1529)‡ | Poor | 47 (3%) | 0.071 (0.029 to 0.113) |
| | Fair | 161 (11%) | |
| | Good | 284 (19%) | |
| | Very good | 449 (29%) | |
| | Excellent | 563 (37%) | |
| Nurse being interested in you as a whole person (n=1295)‡ | Poor | 22 (2%) | 0.027 (0.002 to 0.052) |
| | Fair | 99 (8%) | |
| | Good | 265 (20%) | |
| | Very good | 390 (30%) | |
| | Excellent | 453 (35%) | |
| Patient agenda | | | |

**Table 4** Continued

| | Response | n(%) or mean (SD) | ICC (95% CI) |
|---|---|---|---|
| In the last 12 months did you discuss what was most important for you in managing your own health? (n=1479)* | Not at all | 259 (18%) | 0.017 (0.000 to 0.036) |
| | Rarely | 251 (17%) | |
| | Sometimes | 520 (35%) | |
| | Always | 449 (30%) | |
| Asked how my long-term condition affects my life (n=1412)† | Almost never/generally not | 706 (50%) | 0.036 (0.008 to 0.064) |
| | Sometimes | 321 (23%) | |
| | Most of time/almost always | 385 (27%) | |
| Care plans | | | |
| Do you have a written care plan? (n=1526) | No/do not know | 1375 (90%) | 0.008 (0.000 to 0.023) |
| | Yes | 151 (10%) | |
| I was given copy of my plan (n=1410)† | Almost never/generally not | 1055 (75%) | 0.023 (0.000 to 0.045) |
| | Sometimes | 131 (9%) | |
| | Most of time/almost always | 224 (16%) | |
| Make a plan that I can do in my daily life (n=1425)† | Almost never/generally not | 829 (58%) | 0.027 (0.003 to 0.052) |
| | Sometimes | 223 (16%) | |
| | Most of time/almost always | 373 (26%) | |

The extent to which current care for people with multimorbidity is patient centred from the perspective of patients.
*Taken from LTC6 measure.
†Taken from PACIC measure.
‡Taken from CARE measure.
GP, general practitioner; ICC, intracluster correlation coefficient; LTC6, Long-Term Conditions 6; PACIC, perception of the quality of of chronic illness care.

characteristics. However, in this study we have used anonymised practice records to compare clinical diagnoses and been able to show that participants have similar characteristics to non-participants. The bulk of the findings in this study about patient-centred aspects of care come from self-report from patients and professionals, and we do not know how these relate to actual delivery of care in these practices. However, a key aim of the intervention is to improve patient experience of care, for which self-report is the optimal assessment method.

**Table 5** Clinicians' views on care for people with multimorbidity (n=154 from 33 practices)

| | Total | ICC (95% CI) |
|---|---|---|
| Patients with multimorbidity have a special need for holistic, patient-centred care | 136 (88%) | 0.000 (0.000 to 0.116) |
| Holistic, patient-centred care is enhanced by continuity of care | 148 (96%) | 0.000 (0.000 to 0.116) |
| Patients being reviewed for a long-term condition should be given a written care plan | 96 (62%) | 0.188 (0.023 to 0.352) |
| Patients' main concerns may be overlooked during review of long-term conditions | 88 (57%) | 0.037 (0.000 to 0.165) |
| Patients with three or more conditions need longer appointments to address all their concerns | 143 (93%) | 0.040 (0.000 to 0.167) |

N (%) of clinicians who agree/strongly agree.
The extent to which current usual care aligns with the 3D model, on the basis of practice policies.
ICC, intracluster correlation coefficient.

As a pragmatic trial, 3D is designed to recruit a population with high external validity by ensuring that practices and patients who participate are representative of the wider population to whom the intervention, if effective, would be provided in real life. The overall response rate among patients invited was 33%. This is likely to be an underestimate of the proportion of eligible patients recruited because some non-responders may not have been eligible. Nevertheless, this recruitment rate is typical of previous studies in UK populations of primary care patients with long-term conditions,[30 31] and may be considered relatively high given that the inclusion criteria for this trial selected elderly patients with multiple illnesses.

Our inclusion criteria were based on patients with 3 or more types of condition from a list of 17 conditions included in the QOF framework, a pay-for-performance scheme. The use of a wider list of conditions may have led to selection of a different group of patients, but we based our selection on QOF conditions because they are prevalent, clinically important and reliably coded.

### Interpretation of the findings and comparisons with the wider literature

We raised three main issues in this paper. First, how do practices and patients in 3D compare to the wider primary care population outside the trial? Although limited by available data, the comparisons suggested that the consenting sample did not differ markedly from the potentially eligible population on measured characteristics, with the largest difference being the proportion with dementia or learning difficulties, which is unsurprising given the nature of the recruitment method. We sought to ensure that our inclusion criteria were as wide as possible, but this study further demonstrates the difficulty of recruiting patients with dementia and learning

**Table 6** Results of practice pro forma at baseline

| Question | Yes N (%) | Comments |
|---|---|---|
| Is it your policy to encourage all patients to see their named general practitioner (GP) whenever possible? | 10 (30) | In most practices, patient request and GP availability determined whether they saw their usual GP. In most of the 10 practices saying 'yes' and in many saying 'no' it was practice policy to fulfil the patient request where possible. However, one practice had a formal personal list system ensuring patients saw their own GP. |
| Is it your policy that every patient with a long-term condition (LTC) has a face-to-face medication review at least once a year? | 25 (76) | This could be with GP, pharmacist or nurse prescriber. |
| Is it your policy that every patient with ≥2 LTCs receives a written care plan? | 1 (3) | Most practices said they used care plans for some conditions (most commonly chronic obstructive pulmonary disease, diabetes, learning disabilities and dementia). Other conditions included severe mental health conditions, rheumatoid arthritis, various cardiovascular conditions and epilepsy. Only one practice said they did not use them for any of the 15 conditions listed and three said they only used them for one condition. What practices understood by 'care plan' varied and some distinguished between care plans and self-management plans suggesting that they saw care plans as information primarily for health professionals. |
| Is it your formal policy to annually screen for depression all patients with ≥2 LTCs who are under regular review? | 12 (36) | For those answering 'yes' we checked if they routinely used a formal measure such as PHQ2 or PHQ9 for their screening and only counted it as 'yes' if they did. |
| Is it your policy to offer combined reviews for some patients with multimorbidity? | 31 (94) | 11 practices were offering fully combined reviews which meant they were preplanned, encompassing all LTCs, and both clinician and patient were aware all conditions were to be reviewed. The other 20 either combined a subset of conditions or tried (as far as time and skills allowed) to combine reviews. The remaining two were conducting separate reviews. |

All answers are reports from the key informant in the practice who was usually a senior administrator or practice manager who could consult with clinical colleagues for answers to some questions. Where possible, when there was ambiguity, answers were clarified by follow-up phone calls.
PHQ, Patient Health Questionnaire.

difficulties within trials. Our findings suggest that to increase inclusion rates of people with these conditions it is important to have strategies to encourage patient participation, and to address the reluctance of some clinicians to even allow them to be invited to participate. Although we cannot be sure that patients agreeing to take part do not differ on other important characteristics, the data do provide some confidence that the results are not based on a highly selected sample, especially in terms of physical health conditions.

The second issue is the levels of illness burden, treatment burden and patient-centred care experienced by patients with multimorbidity. Our recruitment method used a simple method of condition counts which is easy to conduct, but it was unclear whether we would identify patients with high needs. In terms of illness burden, our data suggest a sample with relatively high level of morbidity and need. Patients report an average of seven conditions, and nearly two-thirds report general health that is either 'fair' or 'poor'. Patients were receiving a large number of medications and more than one-third of participants reported anxiety or depression. Examining the baseline data also demonstrates that, consistent

with previous literature, patients with multimorbidity are burdened by the demands placed on them by treatment and expectations of self-management.[32 33] Although there are many qualitative papers on the experience of patients with multimorbidity,[34] more quantitative data are needed. The trial recruitment procedures therefore identified a group of patients with significant burdens of illness and treatment whose characteristics seem well matched to the intervention model, and where many patients exceed minimum requirements of the trial eligibility criteria. Our data also suggest that patients do not receive care which they perceive as patient centred in several important respects, as discussed below.

The third issue raised by this paper is an understanding of 'usual primary care' for multimorbidity in this population, to better understand current practice against which the potential benefits of 3D are being assessed. Assessing the 'nature of current care' for multimorbidity, and the degree to which it is 'patient-centred care' is a complex task. Nevertheless, several important findings can be highlighted, linked to the 3D model (figure 1).

Most patients reported satisfaction in general with their care. These high ratings are in line with wider work on

patient perceptions of primary care and might indicate limited scope for improvement, but interpretation of such satisfaction scores is not always straightforward.[35] However, when considering the more structured aspects of care for long-term conditions (as assessed in models such as the Chronic Care Model[36]), the results showed more room for improvement. Many patients reported that their care was not always joined up and although three-quarters of patients in this study had a preferred GP, only 59% reported that they usually consult them. The 3D model identifies eliciting and responding to the patient agenda (their own individual priorities) as a key gap in current care, and the questions from the LTC6 questionnaire and the PACIC scale showed only modest levels of agreement about items relating to this facet of care. This is in line with previous work in a broader population of patients.[37] Similarly, despite a very significant policy focus on care plans,[38] many practices did not have a policy to provide them and most patients did not report receiving them.

Many of the processes of care where we identified gaps (such as improving continuity and coordination of care, establishing the patient agenda to improve shared decision-making, production of care plans,) are a focus of the 3D model. If these processes are mediators of improvements in quality of life, as hypothesised by the logic model underlying the 3D approach, the trial may have a reasonable chance of seeing change in the intended primary and secondary outcomes, assuming it can be implemented.

## Summary

The data suggest our pragmatic trial has achieved reasonable levels of external validity, and that the results should be generalisable to primary care in the UK. Although patients were generally satisfied with their relationships with primary care professionals, there remains significant room for improvement in important aspects of care for multimorbidity that are targeted by the 3D intervention. The pragmatic 3D randomised controlled trial will both test whether our intervention can generate enhancements in those processes of care, and whether those enhancements translate to better patient quality of life, patient experience and value for money.

**Author affiliations**
¹Centre for Academic Primary Care, Population Health Sciences, Bristol Medical School, University of Bristol, Bristol, UK
²NIHR School for Primary Care Research, Centre for Primary Care, Division of Population of Health, Health Services Research and Primary Care, Manchester Academic Health Science Centre, University of Manchester, Manchester, UK
³Bristol Randomised Trials Collaboration (BRTC), Population Health Sciences, Bristol Medical School, University of Bristol, Bristol, UK
⁴Population Health Sciences Division, School of Medicine, University of Dundee, Dundee, UK
⁵Institute of Health and Wellbeing, General Practice and Primary Care, University of Glasgow, Glasgow, UK
⁶Clinical Innovation and Research, Royal College of General Practitioners, London, UK

**Acknowledgements** The authors would like to thank Bristol Clinical Commissioning Group (CCG) for hosting this research, in particular Emma Moody, Joanne Atkinson and Rebecca Robinson. We thank PRIMIS for developing the search. We would also like to thank members of the independent TSC, DMC, advisory group, and public and patient involvement group for their advice and input into the design and conduct of the study. Finally, we would like to thank the practices and patients and trainers for their participation.

**Contributors** CS conceived the original study. CS, PB, SWM, BG, IR, STB, ARGS and CM are coapplicants on the funding application. KC along with PB and CS led the writing of the first draft of the paper with contribution from M-SM (trial manager) and DG (statistical analyses). All authors contributed to the development and editing of this manuscript.

**Funding** This project was funded by the National Institute for Health Research Health Services and Delivery Research Programme (project number 12/130/15). This study was designed and conducted in collaboration with the Bristol Randomised Trials Collaboration (BRTC), a UKCRC registered clinical trials unit (CTU) in receipt of National Institute for Health Research CTU support funding. CS is partly supported by The National Institute for Health Research Collaboration for Leadership in Applied Health Research and Care West. The trial sponsor is the University of Bristol, (Senate House, Tyndall Avenue, Bristol BS8 1TH, UK).

**Disclaimer** The views and opinions expressed therein are those of the authors and do not necessarily reflect those of the HS&DR Programme, NIHR, NHS or the Department of Health.

**Competing interests** None declared.

**Patient consent** Not required.

**Ethics approval** South-West (Frenchay) NHS Research Ethics Committee.

**Provenance and peer review** Not commissioned; externally peer reviewed.

**Data sharing statement** Once the main results have been published, data may be available to other investigators subject to agreement about the protocol with the chief investigator and compliance with policies of the funder and sponsor in relation to data sharing.

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
