## [Reviewer comments · BMJ Open]

ARTICLE DETAILS

TITLE (PROVISIONAL)	Understanding usual care for patients with multimorbidity: baseline data from a cluster randomised trial of the 3D intervention in primary care
AUTHORS	Chaplin, Katherine; Bower, Peter; Man, Mei-See; Brookes, Sara; Gaunt, Daisy; Guthrie, Bruce; Mann, Cindy; Mercer, Stewart; Rafi, Imran; Shaw, Ali; Salisbury, Chris

VERSION 1 – REVIEW

REVIEWER	Alexandra Prados-Torres Aragon Health Sciences Institute (IACS). IIS Aragon.
REVIEW RETURNED	03-Nov-2017

GENERAL COMMENTS	Understanding usual care for patients with multimorbidity: baseline data from a cluster randomised trial of the 3D intervention in primary care This paper addresses an important issue as is the need for a holistic patient-centred approach when caring for patients with multimorbidity, describing the characteristic of practices and patients included in the 3D RCT compared to those non-participating. The existence of a hypothesis to be tested would help to better understand the rationale under this study. As a suggestion, in the introduction section, a reference to other multimorbidity care models recently developed, as it is the Multimorbidity Care Model (Palmer et al., 2017; https://doi.org/10.1016/j.healthpol.2017.09.006) or the Ariadne Principles (Muth et al., 2014; http://www.biomedcentral.com/1741-7015/12/223), and a reference to other ongoing RCT for multimorbid patients (e.g., Multi-PAP RCT; ClinicalTrials.gov ID: NCT02866799) would increase the state of the art of this paper. Regarding the method section, it is needed to make a further effort to better clarify the information reported in this paper, specifying which one belongs to providers and which one to patients, as well as the specific questionnaires and questions addressed in this paper. Additionally, the purpose-designed questionnaire to GPs and practice nurses about beliefs and attitudes regarding care of patients with multimorbidity should be provided as an annex to ensure potential repeatability of the study. The total number of practices potentially sampled must be provided
--

	in the text to offer a global idea of the representativeness of the study. In the Summary sections, those aspects identified in which there is potential room for improvement of care should be specified in order to guide future studies and development of models of care.
--	---

REVIEWER	Andrew W Murphy NUI Galway, Ireland
REVIEW RETURNED	05-Jan-2018

GENERAL COMMENTS	This is a well written paper addressing an important and common topic – namely, how best to manage, in general practice, patients with multimorbidity. It specifically addresses the important, but often overlooked, methodological considerations of external validity and usual care; so indeed I think will be of interest well beyond the specific clinical condition. I have no major comments and will simply provide minor suggestions as they chronologically arise in the manuscript: Page 4: Abstract Objectives: The final paragraph of the introduction provides a very clear and succinct rationale for this paper referring to the key areas of external validity and description of usual care. I believe that by not referring to these constructs in the objectives and indeed the paper title, the merits of the work are being 'undersold'. Results: Significant and important differences between those excluded and those invited are noted in Table 1. It seems somewhat remiss not to refer to these also in this section. Pg 8: Methods Design: "Up to 150 potentially eligible patients were randomly selected from each practice and screened by their GPs". Readers, from a data protection perspective, may be interested as to how this was actually done, and by whom, in practices. Patient data: What does LTC6 refer to ? Staff perceptions: I am not convinced that staff perceptions of care add much to this paper – they are not central to either external validity or description of usual care. They are of interest, but perhaps not for this paper; Table 6 well describes 'usual care'. Table 5 and accompanying text could therefore, in my view, be excluded. Pg 10: Results "Potential participants who were excluded by their GPs were much more likely to have dementia or learning difficulties and less likely to have diabetes or respiratory conditions than those not excluded (Table 2)." This is a really interesting finding, especially when considering that 388 of the 575 excluded, were excluded for reasons other than those explicitly stated. Further discussion may be appropriate later in the paper. Page 11 "However, only 37% reported that their care was always 'joined up'." Was a working definition given to patients as to what was meant by this? Page 12: Discussion
---

	“However in this study we have used anonymised practice records to compare clinical diagnoses and been able to show that participants have similar characteristics to non-participants.” This, and the comparative practice data in Table 1, is a key treasure of the UK primary care system affording real perspective to this paper. Page 17: Figure 2 The actual numbers sampled (5,253) and consented (1,546) differ somewhat from the original sample size numbers of 3,546 and 1,382 respectively (Man SM, 2016). Some comment on this may be appropriate – this is a very minor point. Page 24: References I could not locate reference #15: ‘Worsley SD, Rengerink KO, Irving E, et al. Challenges in Pragmatic Trials: Selection and Inclusion of Usual Care Sites. Journal of Clinical Epidemiology 2017’. Is it In Press ? Well done to all involved.
--	--

REVIEWER	Herzig Lilli Institute of Family Medicine, University of Lausanne, Switzerland
REVIEW RETURNED	09-Feb-2018

GENERAL COMMENTS	Understanding usual car for patients with multimorbidity : baseline data from a cluster randomised trial of the 3D intervention in PC Thank you for allowing me to review this very interesting paper on the baseline data of your CRT. It is not frequent to find CRT about patients themselves in multimorbidity research and this study will surely help to improve some aspects of the management of multimorbid patients in PC. Some minor concerns however : Abstract : Definition of multimorbidity varies greatly between different studies. It would be helpful to have a summary of the precise definition you will use (2 or 3 long term conditions and what kind of list of included long term conditions) Introduction : Well described and clear introduction. However in the second paragraph there is a lack of references for some of the proposal. And it seems necessary to specify some of the most important characteristics of the health system in the UK, to help understanding some of the method items. Method : A general comment : quite a lot of the information can only be found in the additional very long protocol but not in the paper. Therefore it seems necessary to have some summaries of the main information in the paper. Design : Please specify why you use 3 (and not 2) long-term conditions ?
--

	Please specify also when and how the consent of participation patients was given ? Only 11 long-term conditions are included, of course the most prevalent. However low prevalent and rare long-term conditions are the norm and not the exception in PC. A limited list is not really representative of the daily practice. This should be mentioned in the limitation or in the discussion. Patients data : If most of the measures are well described in literature, it would be helpful to have some more informations about them in the paper (a summary as specified above) : what is measured, what are the main items ? And there is no reference for the MTBQ ; at least Tran who developed the TBQ should be mentioned) Is the MTBQ the same as the TBQ ? Same comment for QIPP program – is there a reference ? Usual care is mentioned but for non English reader, it is not so clear what usual care is in UK. Some precision about that may be added in the introduction – see also comment above. In the « staff perception » a questionnaire with 12 statements is mentioned, but a summary description should be added. Results : Well described. However table 1 is difficult to interpret for non UK readers (same comment as above) : a summary description of a usual practice in UK is necessary (organisation ? Is it usual to have a nurse ? How many GPs are working there ? Full time ? Are patient addressed to a PC practice or do they have a choice ?) These informations can be addressed in the introduction. Discussion : No specific comment Additional comment : I didn't found the trial registration nor the checklist
--	---

REVIEWER	François Schellevis NIVEL (Netherlands Institute for Health Services Research), Utrecht, and Department of General Practice & Elderly Care Medicine, Amsterdam Public Health Research Institute, VU University Medical Centre, Amsterdam, the Netherlands
REVIEW RETURNED	14-Feb-2018

GENERAL COMMENTS	Manuscript bmjopen-2017-019845 "Understanding usual care for patients with multimorbidity: baseline data from a cluster randomised trial of the 3D intervention in primary care" This manuscript describes the recruitment procedure of practices, baseline characteristics of participating patients in the 3D trial, and the care provided and received before the start of the 3D intervention. The paper is well written and adds to the existing knowledge, especially regarding the 'usual care' for multimorbid patients in primary care.
--

	I have two remarks about this paper. (1) The authors did not sufficiently take into account the clustering of the data when analyzing the patients' experiences. In the design paper of the 3D trial, the hierarchical nature of the study design is well addressed, but this is also applicable to this paper. For example, satisfaction with current care, treatment burden, care coordination, and assessment of 'whole person care' may all be dependent on preferences or habits of GPs or practices, and can therefore not be considered as statistically independent. "Having a written care plan" is the only variable about which clustering is described. More information should be provided on the clustering of the patients' experiences, e.g. by aggregating scores on GP and/or practice level and presenting ranges of scores between GPs/practices, or – after analyses – by presenting the conclusion of the absence of a clustering effect. Also, I would be interested whether all GPs excluded approximately the same number of patients as being ineligible or whether some GPs excluded many more patients than others. (2) The focus of the first part of the analyses is on comparing participants with non-participants, both on practice and patient level. This is of course important and valid. However, did the authors consider to also compare practices' and patients' characteristics between the allocated study groups (intervention/usual care)? This would inform the readers about the comparability of the practices and participants in the two arms of the trial and thereby add to the value of the paper, especially as starting point for the analyses of the trial outcomes.
--	--

VERSION 1 – AUTHOR RESPONSE

Reviewer(s)' Comments to Author:

Reviewer: 1

Reviewer Name: Alexandra Prados-Torres

Institution and Country: Aragon Health Sciences Institute (IACS). IIS Aragon.

Please state any competing interests or state 'None declared': None declared

Please leave your comments for the authors below

Congratulations, this is a needed and well performed paper. Please, see minor comments in the attach file (BMJ Open 2017.docx) with the objective to increase its overall quality.

This paper addresses an important issue as is the need for a holistic patient-centred approach when caring for patients with multimorbidity, describing the characteristic of practices and patients included in the 3D RCT compared to those non-participating. The existence of a hypothesis to be tested would help to better understand the rationale under this study.

We feel that the rationale for the study is clear from the introduction, and the other reviewers seem to feel it is clear, so we have not changed this.

As a suggestion, in the introduction section, a reference to other multimorbidity care models recently developed, as it is the Multimorbidity Care Model (Palmer et al., 2017; <https://doi.org/10.1016/j.healthpol.2017.09.006>) or the Ariadne Principles (Muth et al., 2014; <http://www.biomedcentral.com/1741-7015/12/223>), and a reference to other ongoing RCT for multimorbid patients (e.g., Multi-PAP RCT; ClinicalTrials.gov ID: NCT02866799) would increase the state of the art of this paper.

We have included reference to the papers by Palmer and Muth. There are a number of ongoing trials in multimorbidity so it would seem inappropriate to just mention one of them.

Regarding the method section, it is needed to make a further effort to better clarify the information reported in this paper, specifying which one belongs to providers and which one to patients, as well as the specific questionnaires and questions addressed in this paper.

We think the sub-headings make clear which information came from patients and which from staff in practices. The patient questionnaire used validated measures which have now all been referenced. We have attached the staff questionnaire as an appendix. In this paper (Table 5) we have just presented data from the questions which relate to the questions of interest for this paper and which can be compared with patients perceptions.

Additionally, the purpose-designed questionnaire to GPs and practice nurses about beliefs and attitudes regarding care of patients with multimorbidity should be provided as an annex to ensure potential repeatability of the study.

See above

The total number of practices potentially sampled must be provided in the text to offer a global idea of the representativeness of the study.

In table 1 we have shown the number of participating and non-participating practices in each area. These data are provided to show the representativeness of participating practices in relation to their local area. However we cannot provide a reliable denominator for the number of practices invited to participate, since practices were recruited in various ways in different areas. Some received information about the trial through a newsletter informing research-active practices about studies open to recruitment, others were approached directly by the research team and some heard about the study through word of mouth and expressed interest. We can report that 68 practices expressed interest in the study, of which 35 agreed to participate but 2 then withdrew prior to randomisation. We have added these data.

In the Summary sections, those aspects identified in which there is potential room for improvement of care should be specified in order to guide future studies and development of models of care.

We have added a sentence to this effect

Reviewer: 2

Reviewer Name: Andrew W Murphy

Institution and Country: NUI Galway, Ireland

Please state any competing interests or state 'None declared': Peter Bower is a co-applicant on a current national grant which I lead from the Irish Health Research Board.

Please leave your comments for the authors below

This is a well written paper addressing an important and common topic – namely, how best to

manage, in general practice, patients with multimorbidity. It specifically addresses the important, but often overlooked, methodological considerations of external validity and usual care; so indeed I think will be of interest well beyond the specific clinical condition.

I have no major comments and will simply provide minor suggestions as they chronologically arise in the manuscript:

Page 4: Abstract

Objectives: The final paragraph of the introduction provides a very clear and succinct rationale for this paper referring to the key areas of external validity and description of usual care. I believe that by not referring to these constructs in the objectives and indeed the paper title, the merits of the work are being 'undersold'.

Thank you. We have added an additional sentence in the abstract.

Results: Significant and important differences between those excluded and those invited are noted in Table 1. It seems somewhat remiss not to refer to these also in this section.

We think the reviewer is responding to some of the small p values and statistical significance. Because of the large sample size, some differences are statistically significant even though they are very small and not meaningful. We have added a sentence to the methods to explain that we have paid more attention to whether differences are meaningful, rather than p values alone, because of the large sample size. We have also added to the abstract the finding that patients with dementia or learning difficulties were more likely to be excluded from the study by their GPs.

Along with the other changes requested this has taken us slightly over the word count for the abstract, and we are not sure if you have some flexibility over this.

Pg 8: Methods

Design: "Up to 150 potentially eligible patients were randomly selected from each practice and screened by their GPs". Readers, from a data protection perspective, may be interested as to how this was actually done, and by whom, in practices.

We have added a sentence about this in the Design section of the methods

Patient data: What does LTC6 refer to ?

We have added a reference to this questionnaire on page 7

Staff perceptions: I am not convinced that staff perceptions of care add much to this paper – they are not central to either external validity or description of usual care. They are of interest, but perhaps not for this paper; Table 6 well describes 'usual care'. Table 5 and accompanying text could therefore, in my view, be excluded.

We have considered this view, but we think the inclusion of the clinicians' views is interesting and relevant. These data highlight that practice staff agreed with the importance of certain aspects of care for multimorbidity targeted by the intervention, yet the responses from patients' in the same practices show that patients did not feel they received these aspects of care (e.g. care plans etc).

Pg 10: Results

"Potential participants who were excluded by their GPs were much more likely to have dementia or learning difficulties and less likely to have diabetes or respiratory conditions than those not excluded

(Table 2).” This is a really interesting finding, especially when considering that 388 of the 575 excluded, were excluded for reasons other than those explicitly stated. Further discussion may be appropriate later in the paper.

Thank you. We have added two sentences about this point to the discussion.

Page 11

“However, only 37% reported that their care was always ‘joined up’.” Was a working definition given to patients as to what was meant by this?

This finding comes from one of the LTC6 questions: ‘Do you think the support and care you receive is joined up and working for you?’ We have added the actual wording of the question to table 4.

Page 12: Discussion

“However in this study we have used anonymised practice records to compare clinical diagnoses and been able to show that participants have similar characteristics to non-participants.” This, and the comparative practice data in Table 1, is a key treasure of the UK primary care system affording real perspective to this paper.

Thank you. No response needed.

Page 17: Figure 2

The actual numbers sampled (5,253) and consented (1,546) differ somewhat from the original sample size numbers of 3,546 and 1,382 respectively (Man SM, 2016). Some comment on this may be appropriate – this is a very minor point.

The number of patients recruited was higher than originally intended because of the length of the process of recruiting practices, sampling patients, inviting them, sending reminders and waiting for their responses. This meant that we had to make decisions about the number of patients to invite before we were sure about the recruitment rate or final recruitment numbers. We invited more patients than originally anticipated in our sample size calculation in order to be sure that we would reach our recruitment target and not end up with an under-powered study.

Page 24: References

I could not locate reference #15: ‘Worsley SD, Rengerink KO, Irving E, et al. Challenges in Pragmatic Trials: Selection and Inclusion of Usual Care Sites. Journal of Clinical Epidemiology 2017’. Is it In Press?

This reference has been updated

Well done to all involved.

Thank you

Reviewer: 3

Reviewer Name: Herzig Lilli

Institution and Country: Institute of Family Medicine, University of Lausanne, Switzerland

Please state any competing interests or state 'None declared': none declared

Please leave your comments for the authors below

Understanding usual car for patients with multimorbidity : baseline data from a cluster randomised trial of the 3D intervention in PC

Thank you for allowing me to review this very interesting paper on the baseline data of your CRT. It is not frequent to find CRT about patients themselves in multimorbidity research and this study will surely help to improve some aspects of the management of multimorbid patients in PC.

Some minor concerns however :

Abstract :

Definition of multimorbidity varies greatly between different studies. It would be helpful to have a summary of the precise definition you will use (2 or 3 long term conditions and what kind of list of included long term conditions)

This information is shown in the Design section of the methods section on page 7, and the specific conditions are listed in Appendix A.

Introduction :

Well described and clear introduction. However in the second paragraph there is a lack of references for some of the proposal.

We have included some additional references to this paragraph

And it seems necessary to specify some of the most important characteristics of the health system in the UK, to help understanding some of the method items.

We have added a few details about this under the 'Design' section of the methods.

Method :

A general comment : quite a lot of the information can only be found in the additional very long protocol but not in the paper. Therefore it seems necessary to have some summaries of the main information in the paper.

Design :

Please specify why you use 3 (and not 2) long-term conditions ?

We have added a sentence to explain this.

Please specify also when and how the consent of participation patients was given?

We have added a few details about this under the 'Design' section of the methods.

Only 11 long-term conditions are included, of course the most prevalent. However low prevalent and rare long-term conditions are the norm and not the exception in PC. A limited list is not really representative of the daily practice. This should be mentioned in the limitation or in the discussion.

We actually included 17 conditions, grouped into 11 types of condition (for example, different cardiovascular conditions were grouped so that if patients had both hypertension and heart failure this only counted once). We recognise that a different list of conditions could have led to selection of a different group of patients, but the QOF list includes the most common and

most clinically important conditions. We have added a section about this to the discussion, under limitations.

We collected details of a longer list of self-reported conditions amongst participants using the Bayliss questionnaire, and as shown in Table 3 participants had a mean of 7.5 conditions from this list.

Patients data :

If most of the measures are well described in literature, it would be helpful to have some more informations about them in the paper (a summary as specified above) : what is measured, what are the main items ? And there is no reference for the MTBQ ; at least Tran who developed the TBQ should be mentioned) Is the MTBQ the same as the TBQ ? Same comment for QIPP program – is there a reference ?

The MTBQ is not the same as the TBQ. We have added references for the MTBQ and the LTC6 questionnaire (from the QIPP programme).

Usual care is mentioned but for non English reader, it is not so clear what usual care is in UK. Some precision about that may be added in the introduction – see also comment above.

We have added some information about this to the design section of the method.

In the « staff perception » a questionnaire with 12 statements is mentioned, but a summary description should be added.

We have added the questionnaire as an appendix and an explanation in the text that we have presented data on items from this questionnaire which can be compared with patients perspectives on the same issues.

Results :

Well described. However table 1 is difficult to interpret for non UK readers (same comment as above) : a summary description of a usual practice in UK is necessary (organisation ? Is it usual to have a nurse ? How many GPs are working there ? Full time ? Are patient addressed to a PC practice or do they have a choice ?) These informations can be addressed in the introduction.

We have added some information about this to the design section of the method, and also improved the description of the data sources for quality of care and patient satisfaction under 'Analysis'.

Discussion :

No specific comment

Additional comment :

I didn't found the trial registration nor the checklist

Trial registration added to abstract. Checklist also attached

Reviewer: 4

Reviewer Name: François Schellevis

Institution and Country: NIVEL (Netherlands Institute for Health Services Research), Utrecht, and Department of General Practice & Elderly Care Medicine, Amsterdam Public Health Research Institute, VU University Medical Centre, Amsterdam, the Netherlands

Please state any competing interests or state 'None declared': None declared

Please leave your comments for the authors below
See attached file (Manuscript bmjopen 2017-019845.docx)

This manuscript describes the recruitment procedure of practices, baseline characteristics of participating patients in the 3D trial, and the care provided and received before the start of the 3D intervention.

The paper is well written and adds to the existing knowledge, especially regarding the 'usual care' for multimorbid patients in primary care.

I have two remarks about this paper.

(1) The authors did not sufficiently take into account the clustering of the data when analyzing the patients' experiences. In the design paper of the 3D trial, the hierarchical nature of the study design is well addressed, but this is also applicable to this paper. For example, satisfaction with current care, treatment burden, care coordination, and assessment of 'whole person care' may all be dependent on preferences or habits of GPs or practices, and can therefore not be considered as statistically independent. "Having a written care plan" is the only variable about which clustering is described. More information should be provided on the clustering of the patients' experiences, e.g. by aggregating scores on GP and/or practice level and presenting ranges of scores between GPs/practices, or – after analyses – by presenting the conclusion of the absence of a clustering effect.

Thank you for this comment and we have given further thought to how best to deal with clustering within these analyses. Table 2 is the only one which describes any statistical comparison between groups (comparing participants and non-participants). These have now been analyzed in multi-level regression models which included practice as a random effect. For tables 3, 4 and 5 we have now added details of the intraclass correlation coefficient to illustrate the extent to which each variable is clustered by practice, and we have commented on the findings in the results.

Also, I would be interested whether all GPs excluded approximately the same number of patients as being ineligible or whether some GPs excluded many more patients than others.

We are able to provide these data at practice level but not at the level of individual GP. We have added a sentence to show that the percentage of eligible patients excluded per practice had a mean of 11.01% and a s.d. 8.01%.

(2) The focus of the first part of the analyses is on comparing participants with non-participants, both on practice and patient level. This is of course important and valid. However, did the authors consider to also compare practices' and patients' characteristics between the allocated study groups (intervention/usual care)? This would inform the readers about the comparability of the practices and participants in the two arms of the trial and thereby add to the value of the paper, especially as starting point for the analyses of the trial outcomes.

In line with CONSORT guidance, a comparison of the characteristics of practices and patients between trial arms will be Table 1 in the paper describing the main trial results. It would not be appropriate to report this between-trial-arm comparison here, since the focus of this paper is establishing the external generalisability of the overall trial population. It would also cause problems of prior publication when we publish the main trial results.

VERSION 2 – REVIEW

REVIEWER	François Schellevis NIVEL (Netherlands Institute for Health Services Research), the Netherlands
REVIEW RETURNED	31-May-2018

GENERAL COMMENTS	I would like to thank the authors for their reply and revisions, which adequately addressed my remarks.
--

REVIEWER	Herzig Lilli Institute of family medicine, university of Lausanne Switzerland
REVIEW RETURNED	01-Jun-2018

GENERAL COMMENTS	Well done, thank you for this good revision Last very minor comment: orthographic mistake in the title of table 6
--